# Simulation of the Human Myopic Eye Cornea Compensation Based on the Analysis of Aberrometric Data

**Pavel A. Khorin** [1,2,*] and **Svetlana N. Khonina** [1,2]

1. Samara National Research University, Samara 443086, Russia
2. Image Processing Systems Institute of RAS—Branch of the FSRC "Crystallography and Photonics" RAS, Samara 443001, Russia
* Correspondence: khorin.pa@ssau.ru or paul.95.de@gmail.com

**Abstract:** Various diffractive, refractive and holographic optical elements, such as diffraction gratings; microlens raster; phase plates; multi-order diffractive optical elements; adaptive mirrors; diffractive and refractive axicons; holographic multiplexes and many others are used to analyze wavefront aberrations. We shortly discuss the features (advantages and disadvantages) of various wavefront aberration sensors in the Introduction. The main part of the paper is devoted to the analysis of the weight coefficients of Zernike polynomials obtained during medical examinations of the cornea in the human eye. Using data obtained by aberrometers, the average values of the Zernike polynomial coefficients for the anterior and posterior surfaces of the healthy eye cornea and a myopic one were calculated. The original wavefront for the anterior and posterior surfaces of the cornea was restored separately, as well as the total wave aberration. For an objective assessment of the quality of vision, the corresponding point spread functions (PSFs) were calculated. We propose to compensate for the aberrations of the myopic eye, taking into account the physical features of the corneal surface. The results of numerical simulation showed that in order to improve the quality of the patient's vision, it is necessary to take into account high-order aberrations of the anterior surface of the cornea in the form of a coma of the third order and aberrations of the fourth order.

**Keywords:** wavefront aberration; Zernike polynomials; myopic eye cornea; numerical simulation

## 1. Introduction

Detection, identification and compensation of wavefront aberrations are in demand in various applications, including vision correction. Not being able to directly measure the wavefront of the light field on the retina of the eye, it is necessary to determine it indirectly, including by measuring the intensity of the light field in a certain plane.

The wavefront sensor is one of the main elements of the adaptive vision correction system. Its task is to measure the aberrations of the wavefront and transmit the results of these measurements to the processing device. The main causes of wavefront aberrations in the eye are the shape and optical properties of the cornea, pupil and lens. In modern diagnostic devices, wave aberrations are described in terms of Zernike polynomials (OSA and ANSI standards). Nowadays, there are a wide variety of wavefront sensors.

### 1.1. Wavefront Sensors

For example, the wavefront of a light field can be reconstructed from an interferogram. This method was proposed as early as 1800 (Fizo, Jamin, Michelson, Jung) [1,2]. It still has unsurpassed accuracy and makes it possible to directly obtain a map of wavefront deviations at very large aperture sizes. The accuracy of interferometers, especially heterodyne ones, exceeds $\lambda/100$. In addition, taking into account the use of data mining and neural networks [3,4], the wavefront of a light field can be reconstructed from an interferogram with a reference beam of a given shape using both a diffractive and refractive optical

element (in particular, **a diffraction grating** for forming a linear interferogram, **a lens** for spherical, **axicon** for conical) [5,6]. Disadvantages of interferometry are well known-they include the sensitivity of the measuring equipment to vibrations, as well as the need for the physical presence of a reference wavefront. In addition, interferometers are able to determine the phase with an uncertainty of $2\pi$, which imposes additional restrictions on the magnitude of detected aberrations.

Hartmann's method [7], which appeared 100 years later, differs in that wavefront deviations are calculated from a set of **sub-apertures.** It covers the full size of the area to be studied with a certain step. It was first described in 1900 by Johann Hartmann. Later it was modified in 1971 by Ronald Shack, and it is used in astronomy to compensate for aberrations in telescopes. The idea of using wavefront technology belongs to J. Bill (1982). A little later, a technology was developed to use aberrometric analysis for **vision** diagnostics. This year, an algorithm for wavefront reconstruction was developed. The Shack-Hartmann wavefront sensor is a device in which the wavefront is divided into separate beams by a matrix of focusing microlenses (**lens raster**) [8–10]. The finite dimensions of each of the sub-apertures lead to restrictions on the magnitude of the detected aberration. Local slopes can only be measured within the area assigned to the microlens. When a focused beam leaves this region, errors in the measurement of slopes occur, leading to phase reconstruction errors [11]. Among the advantages of the Shack-Hartmann sensor, one can distinguish an accuracy comparable to the interference method and achromaticity.

In the 1950s, Fritz Zernike developed a method to visualize the phase of the light field in a direct way. The Zernike phase contrast method [12] is a powerful tool for converting the spatial phase information of an optical beam into a spatial intensity distribution without light absorption. The basic principle is separate a light beam into its Fourier components using a lens and a **phase plate**. The introduced phase shift creates an intensity distribution according to the phase information carried by the higher spatial frequencies. This method has been successfully applied to analyze aberrations and improve resolution in telescopes, in the decoding of phase-coded information, and in the microscopy of biological tissues [13–15]. However, the phase reconstruction is carried out incorrectly, with an increase in the level of aberration due to the limitation of the linear approximation of the expansion of the wavefront in a Taylor series.

**Adaptive** methods are the most versatile tool for wavefront control and correction of optical aberrations over a wide range. The idea of using adaptive optics to compensate for distortions caused by low visibility was first proposed in 1953 by Horace Babcock, and the method of wavefront correction by a compound mirror was proposed and described by V.P. Linnik in 1957 [16,17]. However, the technological level for the development of adaptive optics systems in the 1950s was not yet high enough. The possibility of creating such a system has appeared since the 1980s due to the development of technology and the possibility of computer control and monitoring with high accuracy. Wavefront sensors based on adaptive methods continue to develop and have found their application in such fields as improving the imaging systems of optical microscopes and telescopes, remote sensing of the Earth, clinical research, etc. [18–21]. This approach uses adaptive optics to compensate for distortion, such as a composite or **adaptive mirror**. Among the shortcomings of this method, one can single out the need to use long-converging iterative or optimization algorithms to fully or partially compensate for wavefront aberrations by selecting a complex phase.

In addition, in the 1990s scientific school of Academician V.A. Soifer (V.V. Kotlyar, S.V. Karpeev, S.N. Khonina), a method for detecting wavefront aberrations was proposed. It is based on multichannel diffractive optical elements (DOE) [22–24] that perform in various diffraction orders consistent filtering of phase distributions corresponding to different basis functions. For direct optical measurement of wavefront decomposition coefficients, **multi-order diffractive optical elements** (DOEs) matched to the Zernike function set, which has been successfully applied to wavefront analysis with small aberrations, can be used. Sensors

based on multi-order DOEs provide sensitivity to wavefront deviations no worse than $\lambda/20$, are resistant to vibrations and do not require the use of optical reference elements [24].

Another solution and extension of the adaptive method can be a **multichannel diffractive optical element** matched to phase distributions in the form of Zernike functions. In contrast to the expansion in terms of the Zernike function basis, which provides correct detection of only small aberrations (up to 0.4 wavelength $\lambda$), the proposed approach removes the limitation on the aberration value. The correct detection up to the wavelength $\lambda$ has been confirmed numerically and experimentally) [25,26].

Since the 2000s, continuously developed modifications of methods [27–29] based on the analysis of Zernike polynomials, as well as using digital information processing and data mining. Alternative methods for measuring and reconstructing wavefront aberrations in optical systems, including the human eye, are considered [30–33].

When a significant blurring **of the focal spot** occurs, it makes sense to apply methods focused on the analysis of the intensity distribution pattern formed by an aberrated optical system in one or several planes. In this case, iterative and optimization algorithms are used to reconstruct the phase, as well as machine learning and neural networks [34,35]. The use of neural networks in the problem of recognition of wavefront aberrations is a new, developing approach. However, this approach is problematic at low levels of aberrations, when the pattern is almost indistinguishable from a diffraction-limited focal spot. Therefore, intensity patterns outside the focal plane are often used for analysis, which, in turn, introduces ambiguity into the analysis of aberrations since defocusing is also one of the types of aberrations.

One of the applications **of diffraction axicons** can be used as a sensor of singular beam states and wavefront characteristics. It is possible to improve the detection of spatial anisotropy and the visualization of wave aberrations by supplementing the lens with a diffractive axicon [36]. When the lens is supplemented with an axicon in the plane corresponding to the focal plane of the lens, an out-of-focus picture is formed instead of the focal picture. This allows us to measure out-of-focus patterns and increase the depth of field and its transverse scale without moving the detector device. However, it is necessary to keep a balance between high visualization efficiency and preservation of the characteristic structure of the original scattering function of the aberrated wavefront (the longer the period of the axicon, the higher its efficiency for visualizing aberrations) [37]. Supporting technologies are being developed to provide new types of wavefront sensors based on multilevel diffractive axicons. That brings the advantage of compact optical systems; moreover, multilevel axicons have a higher conversion efficiency compared to binary ones [38].

Another way to detect aberrations is a holographic wavefront sensor, which is based on a **holographic optical element** (HOE). The main difference between a HOE and a DOE lies in their formation. For the HOE, a reference beam is needed, and the DOE is implemented as an amplitude or phase element corresponding to the complex transmission function. The HOE, on which some aberrated wavefront is recorded, can be immediately used for compensation. To use the HOE in this way requires a recording medium. Among the advantages is the need to process information about the value of the radiation intensity only at two points for one aberration. Among the disadvantages-the use of a holographic multiplex (set of HOE) leads to strong and unavoidable crosstalk (intermodal) noise, preventing real use [39].

As an extension of the HOE-based method, Andersen's compromise between the new **holographic sensor** and the traditional Shack–Hartman approach is considered. The resulting device provides information not about the minimum required 10–15 aberrations but about several hundred local deformations of the wavefront, which makes it difficult to process the information received. In addition, in the approach under consideration, the depth of distortion is measured in each "zone" of the wavefront and not at 2 points [40,41].

### 1.2. Aberration Sensors in Ophthalmology

Wavefront sensors (WFS) are the main components in the field of determining the aberrations of the optical system of the human eye for their subsequent compensation. However, none of the designs used and proposed so far provides the simultaneous achievement of high spatial resolution in the pupil of the tested optics and absolute measurement accuracy comparable to that achieved by laser interferometers. The principles of operation of aberrometers can be divided into the Hartmann-Shack method, the ray tracing method, the Cherning principle, etc.

Among the new research over the past 5 years, the following works on the analysis of various types of aberrometers can be distinguished: at the *Moorfields Eye Hospital eye Hospital* (London, UK) analyzed the results of studies of eyes treated under wavefront control using a Peramis pyramidal aberrometer (SCHWIND eye-tech-solutions GmbH) [42]; in the eye clinic *Maja Clinic* (Nis, Serbia) studies were carried out using the WaveLight Allegro Oculyzer, WaveLight Allergo Biograph instruments, DGH Pachette 3 ultrasonic pachymeter [43]; at the *Al-Watani Eye Clinic* (Cairo, Egypt), a study was conducted to detect keratoconus (KC) with higher sensitivity and specificity using Scheimpflug sensors–Oculyzer [44]; at the *Beijing Tongren Eye Center* (Beijing, China), a consistent comparison of Oculyzer and Topolyzer Vario aberrometers was carried out before and during corneal refractive surgery [45]; and at the *Branchevsky Eye Clinic* (Samara, Russia), a comparative analysis of devices based on Plasido, Scheimpflug and OCT for measuring keratometry in patients after laser vision correction was carried out [46].

Among the many ophthalmic measuring devices, there are several most used in the clinical setting and found in the specialized literature. In addition, the presented number of sensors is considered in articles of high-ranking journals, analyzed in theses, and based on the data obtained from these sensors, decisions are made on the diagnosis, as well as surgical intervention.

In practice, new technologies of ocular **pyramidal aberrometers can be used in ophthalmological clinics**. Osiris pyramidal aberrometers provide repeatable and consistent measurements of ocular aberrometry in normal eyes [47]. A wavefront sensor with an expanded source pyramid-shaped optical element has been successfully used to measure aberrations in the human eye. An important advantage of this sensor for the eye is the easy adaptation to variations in the range of aberrations that can be expected in the optics of the human eye: from very slightly aberrated normal eyes to extremely aberrated eyes in patients with pathological corneas. The disadvantage is that the sensor collects light from false reflections from the surfaces of the eye.

One of the classic technologies for analyzing aberrations in the human eye is the **Hartmann-Shack type aberrometer**. For example, using a tracer aberrometer (iTrace) and a Hartmann-Shack aberrometer (Topcon KR-1 W) provide excellent repeatability but less reliable reproducibility when measuring high-order aberrations [48]. Portable wavefront aberrometer (*hand-held wavefront aberrometers*) with postcycloplegic autorefraction (*postcycloplegic autorefraction-AR*) and cycloplegic refraction (*cycloplegic refraction–CR*) showed good agreement between measurements with postcycloplegic AR and CR in spherical equivalents but tended to give results with falsely detected myopia [49].

Another well-known solution to the phase problem in ophthalmology is the **Scheimpflug sensor**. For example, the DRS Analyzer (Galilei; Ziemer Ophthalmology) uses two rotating Scheimpflug cameras in combination with Placido's topographic system. It uses the Placido disk to provide more accurate anterior curvature topographic data, in addition to the data obtained from the Scheimpflug cameras. Overall, the DRS analyzer provides anterior segment measurements with good repeatability and reproducibility for both normal and refractive corneas [50]. For the detection of keratoconus (KC) with higher sensitivity and specificity in ophthalmological offices, aberrometers such as Scheimpflug tomography, for example, Oculyzer (Alcon), are used. The device is intended for performing computed tomography of the cornea and examination of the anterior part of the eyeball.

Existing and other technologies, on the basis of which clinical aberrometers are built, provide fairly accurate measurements of the deflection of the wavefront of the eye. At the same time, high-order aberrations are measured, which makes it possible to evaluate individual deviations of the wavefront, including those associated with professional activity or age-related changes, in order to optimize the optical (contact or intraocular lenses) or surgical correction of the human eye.

## 2. Materials and Methods

The human eye can be described as a lens system consisting of 3 main components: cornea, pupil and lens [51]. Optical elements work in concert to create an image on the retina. Improper functioning of them leads to the appearance of visual defects-from a slight blur of the image to complete blindness.

The cornea (including the tear film) is the dominant structure of the optical power of the eye (up to 70% on average). Accordingly, it is the main source of aberrations in the eye. The anterior surface of the cornea has an elongated profile. The central region is steeper than the periphery. This shape helps reduce spherical aberrations throughout the eye. However, the shape of the cornea varies greatly from person to person, and this leads to astigmatism and high-order asymmetric aberrations (e.g., coma).

In [52], the correspondence of the main canonical aberrations to various pathologies of the cornea (myopia of varying degrees) of the human eye is determined based on the Zernike polynomial coefficients calculated from the results of aberrometry in medical research. We consider a data set obtained on the basis of *Femto Clinic, Excimer Center "Laser Vision Correction"* ("*Branchevsky Eye Clinic*", Samara, Russia). Based on the simulation of the operation of the optical system of the Fourier correlator and the optical system of the human eye in the Zemax environment, an objective assessment of PSF was carried out for various pathologies of the cornea of the human eye.

Based on the data obtained, an analysis of aberrations of the cornea of the human eye was carried out [53]. As a result, the basic Zernike functions were identified, which are most characteristic of some pathologies of the cornea (myopia of various degrees), the compensation of which leads to an improvement in the quality of the formed image. As a result of the analysis, the correspondence of the main canonical aberrations to the pathologies of the cornea of the human eye was determined, and small groups of informative coefficients were identified.

In this paper, we investigate the possibility of compensating wave aberrations of the human myopic eye by adjusting the heights (weight coefficients of Zernike polynomials) of the anterior surface of the cornea. Correction of the anterior surface of the cornea is convenient to carry out during surgical interventions. We propose to compensate for the wavefront aberrations of the myopic eye, taking into account the physical features of the anterior surface of the cornea. We numerically simulate the compensation of the coefficients of aberrations of the anterior surface of the cornea with a deviation of the total wave aberration of the cornea from the reference model up to 30% and perform an objective assessment of the quality of the formed image based on the PSF.

To obtain data on the aberrations of the optical system of the eye, aberrometers were used: WaveLight Oculizer II and HD Analyzer. Wavelight Oculizer II is a diagnostic device designed to examine the eye. It is used to visualize the anterior part of the eye, which includes the cornea, pupil, anterior chamber, and lens of the eye. This device measures the height of the cornea. It is designed specifically for ophthalmologists. The main advantage of the WaveLight Oculizer II analyzer for our research work is that it analyzes the anterior and posterior surfaces of the cornea according to Zernike functions.

Image quality assessment consists in observing the system in the image area and measuring the photometric structure of this image, i.e., the definition of the scattering function (for example, PSF). In practice, exactly these characteristics quantify the image quality of an optical system. Methods for evaluating image quality of this kind have one

great advantage-they take into account all the factors without exception that are involved in the formation of the structure of a real optical image.

The experimentally obtained PSF characterizes the quality of the system. It makes it possible to take into account all the features of the wave surface formed by the optical system, including the nature of the microrelief of optical surfaces, including the posterior and anterior surfaces of the cornea [54].

The data obtained by the recording device HD Analyzer are also used in the work. This is an instrument based on the double pass light technique that provides an objective clinical assessment of the quality of the optics of the eye.

We consider the Zernike functions in the following form [55]:

$$Z_{nm}(r,\varphi) = \sqrt{\frac{n+1}{\pi r_0^2}} R_n^m(r) \begin{Bmatrix} \cos(m\varphi) \\ \sin(m\varphi) \end{Bmatrix}, R_n^m(r) = \sum_{p=0}^{(n-m)/2} \frac{(-1)^p(n-p)!}{p!((n+m)/2-p)!((n-m)/2-p)!} \left(\frac{r}{r_0}\right)^{n-2p} \tag{1}$$

where $R_n^m(r)$ are the radial Zernike polynomials, and $r_0$ is the aperture radius.

Wavefront aberrations occurring in optical systems are usually described in terms of Zernike functions as follows [56]:

$$W(r,\varphi) = \exp[i\psi(r,\varphi)], \psi(r,\varphi) = \sum_{n=0}^{n_0} \sum_{m=-n}^{n} c_{nm} Z_{nm}(r,\varphi). \tag{2}$$

Appendix A presents a fragment of the calculated and averaged values of the Zernike weight coefficients on the anterior and posterior surfaces of the cornea. Based on the data obtained in the eye clinic, in the form of weight coefficients (the first 15 basis functions) of the Zernike polynomials of the anterior $C_{nm}^1$ and posterior $C_{nm}^2$ surfaces of the cornea, the averaged values for a healthy eye were calculated (Figure 1). It should be noted that the weight coefficients of the Zernike polynomials have different signs for most pronounced canonical aberrations on the anterior and posterior surfaces of the cornea, which leads to partial intracorneal compensation of the aberrations [53]. Therefore, there is no need to compensate them by changing the heights of the anterior surface of the cornea. Here and below, the distribution of the coefficients of the total wave aberration is presented without taking into account the basis of Zernike functions with indices $(n, 0)$.

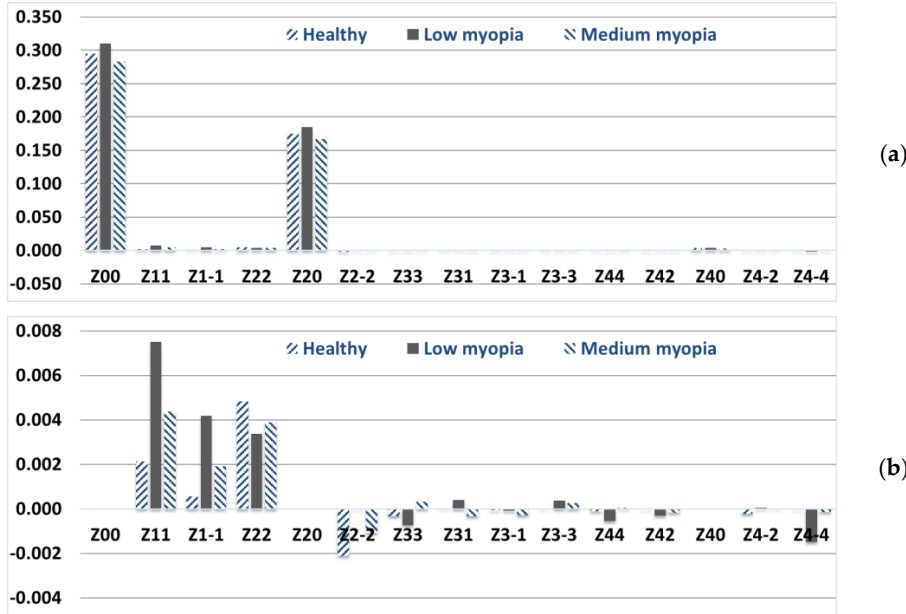

**Figure 1.** Zernike polynomial weight values for total wave aberration in healthy people and people with low/medium corneal myopia: (**a**) first 15 Zernike basis functions, (**b**) Zernike basis functions other than $(n, 0)$.

It can be seen from the distribution of the Zernike polynomial coefficients in Figure 1 that the aberrations corresponding to basis functions other than $(n, 0)$ change depending on the patient's diagnosis. It is proposed to calculate the wavefront and the corresponding PSF for all considered basis functions and different from $(n, 0)$ for the anterior and posterior surface of the cornea and the total wave aberration (Table 1).

**Table 1.** Reconstructed wavefront (WF) and corresponding PSF for a cornea without pathology.

| Zernike Basis Functions | Anterior Surface of the Cornea | | Posterior Surface of the Cornea | | Anterior and Posterior Surfaces of the Cornea | | Image Plane |
|---|---|---|---|---|---|---|---|
| | WF | PSF | WF | PSF | WF | PSF | PSF |
| All |  |  |  |  |  |  |  |
| Without $(n, 0)$ |  |  |  |  |  |  |  |

According to the PSF intensity distribution in the 1st row of Table 1, it can be seen that the cornea without pathology forms an Airy spot with an accuracy of spherical aberration, which advises the physical shape of the cornea. In addition, it is worth noting that there is a mechanism for intraocular aberration compensation due to additional distortions that the pupil and lens bring into the human eye system [57]. There is strong evidence for compensation of aberrations between the cornea and intraocular optics in the case of astigmatism, coma and spherical aberration [58].

As a result of the study of PSF on the anterior and posterior surface of the cornea, it was found that aberrations on the anterior surface of the cornea do not significantly affect the quality of PSF compared to aberrations on the posterior surface of the cornea. It should be noted that laser correction of the cornea is possible only on its anterior surface. Thus, it is necessary to comprehensively assess the level of aberration of the entire cornea $C_{nm}^0 = C_{nm}^1 + C_{nm}^2$ and the anterior and posterior surfaces, and it is the anterior surface that should be "curved".

In addition, for an objective assessment of the quality of PSF on the retina, taking into account healthy intraocular optics (pupil and lens), it is proposed to compensate for wavefront aberrations with healthy cornea aberrations (coefficient values in Figure 1 (healthy)). Therefore, for a healthy cornea, after modeling additional compensation (intraocular compensation), the weight coefficients from Figure 1 (healthy) will register PSF on the retina similar to Airy's spot (Table 1, column "Image plane").

To simulate the compensation of wave aberrations of the human myopic eye cornea, it is proposed to change the weight coefficients of the Zernike polynomials of the anterior surface of the cornea $C_{nm}^1$, which are different from $(n, 0)$ As a reference set (for a healthy eye) of weight coefficients, the average values obtained as a result of measuring the heights of the cornea are considered $C_{nm}^{00} = C_{nm}^{01} + C_{nm}^{02}$. To calculate the coefficients of the total wave aberration $C_{nm}^{\Delta 0}$, which must be compensated, it is required to subtract the vector of Zernike coefficients $C_{nm}^{00}$ for a healthy cornea component by component from the vector of Zernike coefficients $C_{nm}^0$ for a cornea with pathology: $C_{nm}^0 - C_{nm}^{\Delta 0} = C_{nm}^{00}$. Therefore, the values of the Zernike coefficients of the anterior surface of the cornea to be compensated have the following form: $C_{nm}^{\Delta 0} = C_{nm}^{\Delta 1} = C_{nm}^0 - C_{nm}^{00} = (C_{nm}^1 + C_{nm}^2) - C_{nm}^{00}$. Figures 2 and 3 show the distribution of weight coefficients $C_{nm}^{\Delta 0} = C_{nm}^{\Delta 1}$, which must be subtracted from the anterior surface of the cornea for diagnoses of low and medium myopia.

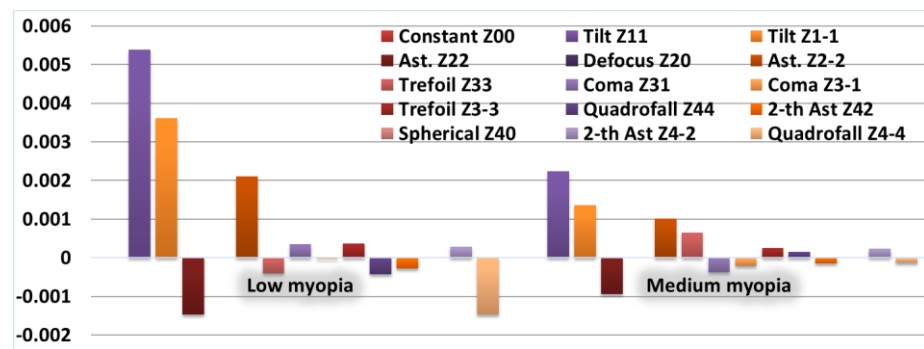

**Figure 2.** Values of the coefficients that must be subtracted from the anterior surface of the cornea for diagnoses of low and medium myopia (presentation by diagnoses).

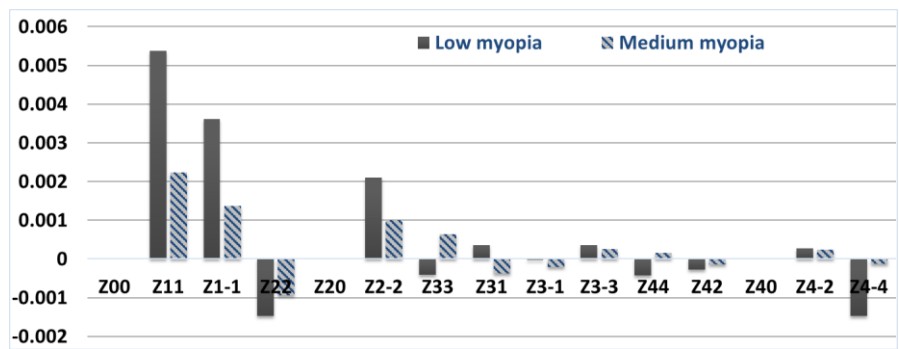

**Figure 3.** The values of the coefficients that must be subtracted from the anterior surface of the cornea for diagnoses of low and medium myopia (through the view).

Table 2 shows the correspondence of the main canonical aberrations to various pathologies (diagnosis of myopia of varying degrees) of the cornea of the human eye based on the coefficients of the Zernike polynomials and the range of diopter values for the sphere and cylinder.

**Table 2.** Correspondence of the diagnosis, range of diopter values for spheres and cylinders, and Zernike polynomials with the largest weight coefficients.

| Diagnosis | | Diopter Range ($D$) for Sphere and Cylinder | | Zernike Polynomials ($n$, $m$) with Largest Weights $C_{nm}^0$ |
| --- | --- | --- | --- | --- |
| | | **Sph, $D$** | **Cyl, $D$** | |
| *Myopia* | *Low* | $[-3;-1]$ | $[-1;0]$ | $(1, \pm1)$; $(2, \pm2)$; $(4,-4)$ |
| | *Medium* | $[-6;-3]$ | $[-1;0]$ | $(1, \pm1)$; $(2, \pm2)$; $(3.3)$; $(3.1)$ |

## 3. Results

### 3.1. Simulation Compensation for Aberrations in Low Myopia Eye

Given the spherical shape of the cornea, we propose to minimize the correction of the weight coefficients of the Zernike polynomials with indices ($n$, 0) and select small groups of coefficients (Table 2). We simulate the compensation of the coefficients of aberrations of the anterior surface of the cornea with the deviation of the total wave aberration of the cornea from the reference model by more than $s$%. For low myopia at $s = 5$% among the compensated polynomials of the anterior surface of the cornea, there will be all Zernike functions except (3, −1) different from ($n$, 0); (1, ±1), (2, ±2), (4, −4) remain at $s = 10$%; (1, ±1), (2, −2) remain at $s = 30$%; just (1, 1) remains at $s = 75$%. The reconstructed wavefront and the corresponding PSF for the cornea with low myopia before and after the simulation of aberration compensation with a deviation from the reference model by more than $s$%

are presented in Table 3. The PSFs on the retina after intraocular compensation are also presented in Table 3. The level of compensation $q$ can be calculated according to the formula $q = 100 - s$.

**Table 3.** Wavefront and corresponding PSF for a cornea with low myopia before and after simulation of aberration compensation with a deviation (RMS) from the reference model by more than $s\%$.

| Anterior Surface of the Cornea | | Posterior Corneal Surface | | Anterior and Posterior Surface of the Cornea | Image Plane | | Compensation Level, $q\%$ |
|---|---|---|---|---|---|---|---|
| WF | PSF | WF | PSF | PSF | PSF | RMS | |
|  |  | | |  |  | 0.0003 | 100% |
|  |  | | |  |  | 0.0010 | 95% |
|  |  |  |  |  |  | 0.0028 | 90% |
|  |  | | |  |  | 0.0081 | 70% |
|  |  | | |  |  | 0.0240 | 25% |
|  |  | | |  |  | 0.0520 | 0% |

According to the PSF distribution on the retina (Table 3, eighth column "Image plane/PSF"), it can be seen that it is enough to compensate for the coefficients of the basis functions that deviate from the reference model by at least $s = 30\%$. In this case, the standard deviation of the ideal PSF on the retina (Table 3, first line, ninth column) from the aberrated one does not exceed 1%, or in absolute terms, is equal to 0.0081. Based on the RMS data, we conclude that compensation for small group aberrations is sufficient to achieve a high quality of the formed image.

Using the HD Analyzer ophthalmological device, which is based on the double-pass light technique and provides an objective assessment of quality eye optics, a study was conducted on the author of this paper (eye without pathologies) in order to obtain a real PSF on the retina and compare with the obtained numerical results. Figure 4a,b shows the report of the diagnostic device based on the results of examining a healthy eye. It can be seen that for the eye without pathologies, PSF on the retina differs from the Airy spot. This is due to the fact that the eye actually forms a non-ideal image, as well as the presence of noise from the recording device during the double pass. In turn, high visual acuity is achieved due to the additional processing of information by the human brain. Figure 4c shows PSF on the retina of other persons without eye pathologies. Thus, we conclude that the obtained PSF on the retina of a healthy eye in a numerical experiment, which differs from the Airy spot, fully corresponds to real PSF. Both types of PSF have one pronounced maximum and an imperfect elongated spot shape.

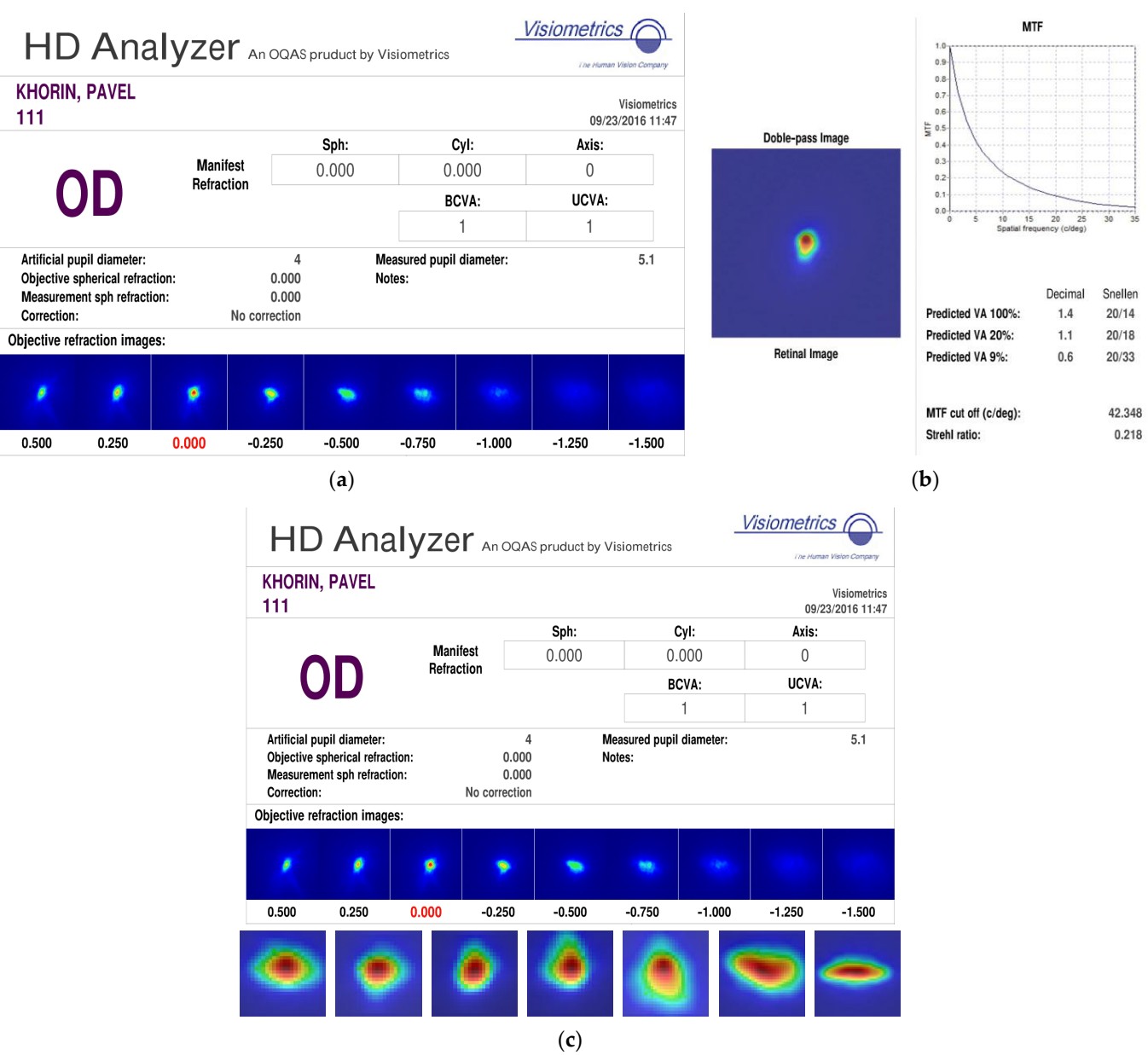

**Figure 4.** Report of the diagnostic device based on the results of a study of a healthy eye (**a**,**b**); PSF on the retina of patients without pathologies (**c**).

### 3.2. Simulation Compensation for Aberrations in Medium Myopia Eye

Similarly, it is possible to simulate partial compensation on the anterior surface of the cornea in the diagnosis of "medium myopia" using the advisory coefficients from Figure 3 and also evaluate the PSF on the retina after intraocular distortion correction (coefficients from Figure 1 (healthy)).

It should be noted that this compensation can be carried out using a WaveLight® EX500 refractive laser or equivalent. WaveLight® EX500 is a stationary excimer spot laser system used in refractive surgery for the treatment of myopia, myopic astigmatism, hyperopia, hyperopic astigmatism, mixed astigmatism, phototherapy keratectomy (PTK) and customized refractive surgery based on data from WaveLight GmbH diagnostic devices or the weight coefficients of the Zernike polynomials.Figure 5 shows the restored WF on the anterior surface before (a) and after (b) laser vision correction, as well as the corresponding spatial corneal thickness profiles (c) obtained using the WaveLight Oculyzer II device for an eye diagnosed with medium myopia.

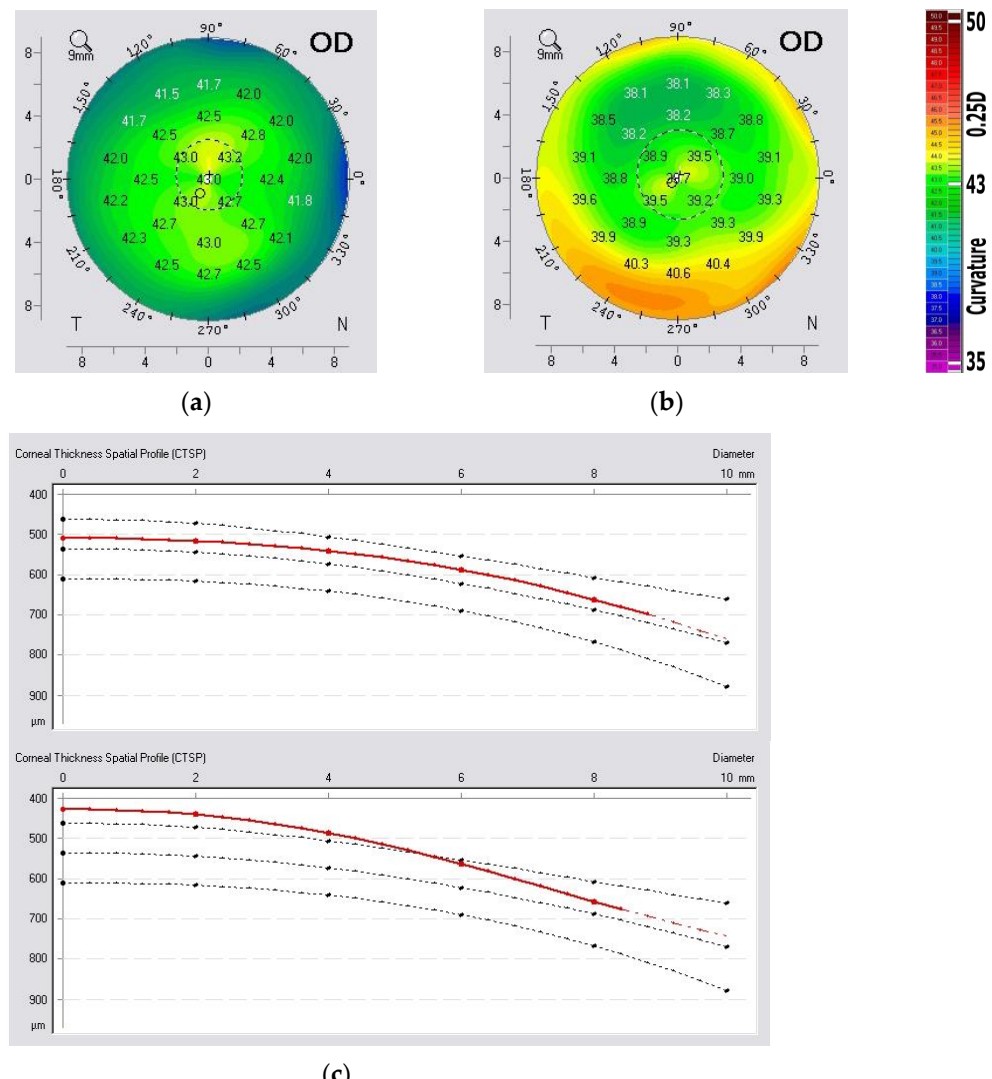

(**a**)   (**b**)

(**c**)

**Figure 5.** Reconstructed WF on the anterior surface cornea before (**a**) and after (**b**) laser vision correction; corresponding spatial profiles of corneal thickness (**c**): red–current, dotted line–ideal.

Figure 5a,b shows that the WF, which is a height map of the anterior surface of the cornea, has changed significantly. Figure 6 shows the weight coefficients of the Zernike polynomials of the total wave aberration before and after laser vision correction, taking into account intraocular compensation (the reference corresponds to zero values of the coefficients).

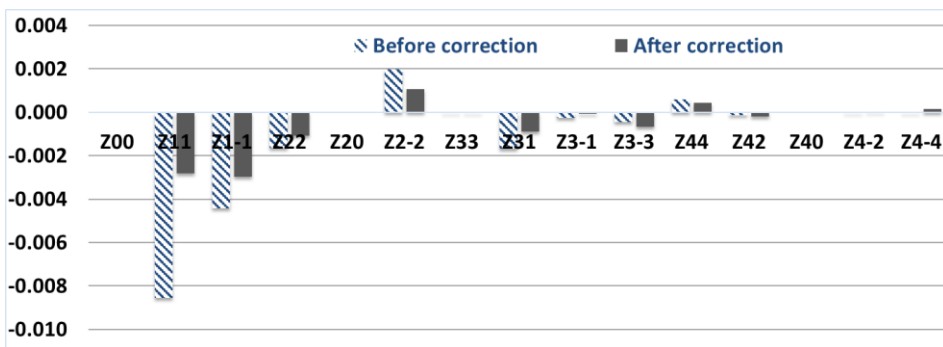

**Figure 6.** Zernike polynomial weight values for total wave aberration before and after correction taking into account intraocular compensation (Zernike basis functions other than (*n*, 0)).

It can be seen from the distribution of coefficients in Figure 6 that, after correction, the Zernike weight coefficients decrease on average, which should lead to an improvement in the quality of the PSF and the patient's vision. This is confirmed by the distribution of intensity on the PSF in Table 4 and by reducing the RMS of the obtained PSF relative to the ideal one (Airy spot).

**Table 4.** PSF simulation on the retina before and after laser vision correction.

| Stage | Reference | Before Correction | After Correction |
| --- | --- | --- | --- |
| PSF in the plane of the formed image |  |  |  |
| RMS | 0 | 0.0190 | 0.0025 |

Table 4 shows that the PSF after correction has a smaller area, and the standard deviation of the reference PSF from the calculated one is equal to 0.0025 (almost 7.5 times less than the standard deviation of the reference PSF relative to the PSF on the retina before correction, equal to 0.0190).

Thus, according to the results of the simulation of aberration compensation (Table 3, ninth column) and laser vision correction (Table 4, third row), the standard deviation from PSF on the retina does not exceed 1% or in absolute values are 0.0028 and 0.0025 respectively. Thus, compensation for aberrations of a small group (low-order aberrations decreased on average by two times, coma-type aberrations (3.1) by 1.8 times and quadrofall (4.4) by 1.4 times, respectively) is sufficient to achieve a high-quality generated image.

## 4. Discussion

The problem of phase recovery has been an urgent problem for more than 200 years. Each of the considered methods for detecting, visualizing, and analyzing wavefront aberrations has its own advantages and disadvantages. To select the most appropriate method in ophthalmology, it is necessary to assess the individual characteristics of the patient, including those associated with professional activities or age-related changes, as well as take an anamnesis. So far, automated diagnosis requires an external observer who, based on a combination of factors and empirical experience, can choose a method or a cascade of methods that best solves a specific problem of assessing a patient's vision for subsequent correction.

In addition, we presented a number of studies of ophthalmic measuring devices which are most used in clinical settings and found in specialized literature. Some of the current research on the analysis of various types of aberrometers in ophthalmic clinics around the world, from Europe to Asia and Africa, are highlighted.

The expansion of wavefront sounding methods has led to a new look at the significance of refractive errors in clinical ophthalmology. Clinical aberrometers provide detailed measurements of the wavefront aberrations of the eye. The distribution and contribution of each higher-order aberration to the total wavefront aberration can now be accurately determined and predicted individually. Note that the measurement of aberrations in the optical structures of the eye is also important for surgery. The possibility of diagnosing aberrations of optical structures during a routine appointment will allow for the correct interpretation of clinical data and reduce the number of errors in the choice of treatment tactics. Increasing the accuracy of the study of aberrations will significantly reduce the classification error, which will lead to an increase in the reliability of the diagnosis when automating the diagnostic process.

Despite the variety of wavefront sensors such as Topolyzer, Oculyzer, Analyzer, Tomey TMS, Pentacam HR, OPD-Scan, Peramis, Allergo Biograph, DGH Pachette, Plasido,

Scheimpflug, Osiris, Topcon KR, iTrace and DRS, none of the designs used or proposed so far provide the simultaneous achievement of high spatial resolution in the pupil of the tested optics and absolute measurement accuracy comparable to that achieved by laser interferometers.

We would also like to point out that not all supporting technologies of diffractive and refractive optics are applied in ophthalmic devices. For example, adaptive methods in combination with data mining and convolutional neural networks could act as a highly effective tool for assessing wavefront aberrations and informing the best way to compensate for them. In addition, diffractive optical elements such as multichannel elements matched with different basis and non-orthogonal functions, axicons, phase plates; holographic optical elements; spatial light modulators could act as supporting technologies for new types of wavefront sensors.

There are two main limitations of the study, which may be addressed in future work. First, the study focused on the 20–40 age group, which does not fully reflect the population. The second limitation is related to the methodology and is associated with the dataset size and the method of measuring the weights of the Zernike polynomials.

The first limitation probably does not significantly affect the results of our study. This conclusion can be justified due to the successful establishment of the relationship between the various weights of the Zernike polynomials and the objective diagnoses of patients in this age group. In the previous study [52], the clustering error for this dataset (150 measurements) using the K-means method is in the range of 0.024–0.049, which does not exceed 5%.

With regard to methodological limitations, the data were measured indirectly. In the first stage, the device measures the relief heights of the anterior from the posterior surface of the cornea at a limited number of points. Then, based on the internal algorithms of the device, it recalculates the relief heights into the values of the weight coefficients of the Zernike polynomials (OSA). It is worth noting that, depending on the number of Zernike basis functions chosen (in the framework of this work, 15 basis functions are considered, i.e., aberrations up to the fourth order inclusive), the values of the weight coefficients of the Zernike polynomials will differ. This limitation should be taken into account when using the results of this work, both in practice for vision correction and in further research work.

Thus, further research should take into account the described limitations of the data collection method and consider alternative solutions. They may consist in using a larger sample of patients, choosing a different method for measuring the weights of the Zernike polynomials, and assessing the influence of the number of basis functions on their weights.

As for the results, we found correspondences of the main canonical aberrations to various pathologies of the human eye based on the coefficients of low- and high-order Zernike polynomials, which can simplify the method for correcting the curvature of the surface of the cornea of the eye, to improve the quality of the formed image and to carry out a more accurate formalization of the diagnosis in terms of optical wave aberrations.

In classical ophthalmology, when correcting the curvature of the surface of the cornea of the eye to improve the quality of the patient's vision, polynomials of the first and second degrees, which are responsible for the so-called "cylinder" and "sphere", are usually compensated. Given the spherical shape of the cornea, it is proposed to minimize the correction of the weight coefficients of the Zernike polynomials with indices ($n$, 0) and select small groups of coefficients.

The results of our research have shown that high-order aberrations turned out to be the most effective for solving a specific problem of compensating for PSF image distortions. An interesting fact is that among them, there are polynomials of the high orders $Z_{31}$ and $Z_{44}$ (above the second order). The conducted research shows that taking into account high-order aberrations (third-order aberrations and fourth-order aberrations) made it possible to significantly improve the quality of the generated image.

It is important to say that the clinical implications of this work may change depending on modern methods of ophthalmic diagnostics and the vision correction process. Consider-

ing the evolving technologies of artificial intelligence, it becomes possible to realize more accurate and personalize compensation and identify the most complex types of aberrations using data mining. What can be provided, both on the basis of surgical intervention and with the help of advanced optical technologies (for example, progressive glasses and personalized contact lenses)?

Note that in order to simulate the compensation of the most pronounced aberrations and, therefore, in order to remove the main influence of the distortions revealed during the expansion of the field in terms of the Zernike basis, it is sufficient to create a field that is complex conjugate to the identified aberration. Such a field can be created by methods of diffractive optics [59], including the application of an appropriate diffractive relief to the lens surface [60], or aberration compensation can be performed using a WaveLight® EX 500 refractive laser and analogs. Thus, it becomes possible to perform customized refractive surgery based on data from diagnostic devices or weight coefficients of Zernike polynomials.

The prospects for further development of the topic are related to the improvement of the developed methods, the enhancement of digital processing algorithms for a complex intensity distribution pattern, and the development of a hardware implementation of a wavefront aberration sensor based on a tunable multichannel element implemented on a spatial light modulator.

## 5. Conclusions

In this paper, groups of canonical aberrations associated with common pathologies of the cornea of the eye (myopia of varying degrees) are identified: for the diagnosis of low myopia, a significant effect of distortion and astigmatism was revealed; for the diagnosis of medium myopia, the influence of third-order aberrations also becomes significant.

A method has been developed to compensate for the aberrations of the myopic eye described by Zernike polynomials, taking into account the possibility of correcting only the anterior surface of the cornea. It is numerically shown that when correcting the identified aberrations of the anterior surface of the cornea for medium myopia, the standard deviation of the aberrated PSF from the reference one (Airy pattern) decreases from 0.019 (which is 2 times higher than for a healthy eye) to 0.0025. The method provides a 7.5-fold improvement in the standard deviation of the aberrated PSF from the reference one (from the Airy pattern) in medium myopia as a result of correcting the shape of only one anterior surface of the cornea.

Using the diagnosis of "low myopia" as an example, it is shown that modeling compensation for aberrations of the anterior surface of the cornea with varying degrees of deviation of the total wave aberration of the cornea from the reference model makes it possible to achieve high-quality PSF in the image plane. It is shown that in order to achieve high quality of the generated image (RMS of the aberrated PSF from the reference should be no more than 1%), compensation of several Zernike polynomials (1, $\pm$1), (2, $\pm$2), (4, $-4$) with the largest deviation is sufficient, rather than the complete set of Zernike basis functions (the first 15 functions). It is shown that the compensation of aberrations of the order (4, $-4$) is important for improving the quality of the PSF. If this recommendation is followed, the standard deviation decreases by almost 3 times (from 0.0081 to 0.0028). In classical ophthalmology, when correcting the curvature of the surface of the cornea of the eye to improve the quality of the patient's vision, polynomials of the first and second degrees, which are responsible for the so-called "cylinder" and "sphere", are usually compensated.

When correcting the vision of the eye with medium myopia, it was found that the PSF after the correction has a smaller area, and the RMS of the reference PSF from the calculated value is 0.0025 (almost 7.5 times less than the RMS of the reference PSF relative to the PSF on the retina before correction, equal to 0.0190).

**Author Contributions:** Conceptualization: S.N.K. and P.A.K.; methodology, S.N.K. and P.A.K.; software, P.A.K.; validation, S.N.K. and P.A.K.; formal analysis, S.N.K. and P.A.K.; investigation, S.N.K. and P.A.K.; resources, S.N.K.; data curation, S.N.K. and P.A.K.; writing—original draft preparation, S.N.K. and P.A.K.; writing—review and editing, S.N.K. and P.A.K.; visualization, S.N.K. and P.A.K.; supervision, S.N.K.; project administration, S.N.K.; funding acquisition, S.N.K. All authors have read and agreed to the published version of the manuscript.

**Funding:** This work was supported by the Russian Science Foundation under Grant No. 22-12-00041 (numerical research) and by the Ministry of Science and Higher Education of the Russian Federation under the FSRC "Crystallography and Photonics" of the Russian Academy of Sciences (the state task No. 007-GZ/Ch3363/26) (comparative analysis).

**Institutional Review Board Statement:** Not applicable.

**Informed Consent Statement:** Not applicable.

**Data Availability Statement:** Data is contained within the article.

**Acknowledgments:** We acknowledge the equal contribution of all the authors.

**Conflicts of Interest:** The authors declare no conflict of interest.

**Appendix A**

A fragment of the values of the Zernike weight coefficients on the anterior and posterior surfaces of the cornea is presented in Tables A1 and A2.

**Table A1.** Values of Zernike Weight Coefficients on the Anterior and Posterior Surface of the Cornea in the Diagnosis of Low Myopia.

| Type | Znm | Cnm | | | | | |
|---|---|---|---|---|---|---|---|
| | | **Anterior Surface** | | | | | |
| Constant | Z00 | $1.30 \times 10^{-1}$ | $1.30 \times 10^{-1}$ | $1.34 \times 10^{-1}$ | $1.33 \times 10^{-1}$ | $1.41 \times 10^{-1}$ | $1.42 \times 10^{-1}$ |
| Tilt | Z11 | $8.32 \times 10^{-4}$ | $2.89 \times 10^{-4}$ | $-4.18 \times 10^{-4}$ | $-1.52 \times 10^{-3}$ | $2.25 \times 10^{-4}$ | $1.82 \times 10^{-3}$ |
| Tilt | Z1−1 | $-8.56 \times 10^{-4}$ | $-5.51 \times 10^{-4}$ | $-1.33 \times 10^{-3}$ | $8.90 \times 10^{-4}$ | $-3.55 \times 10^{-4}$ | $-4.10 \times 10^{-5}$ |
| Astigmatism | Z22 | $1.73 \times 10^{-3}$ | $1.44 \times 10^{-3}$ | $6.35 \times 10^{-4}$ | $6.94 \times 10^{-4}$ | $7.54 \times 10^{-4}$ | $8.81 \times 10^{-4}$ |
| Defocus | Z20 | $7.63 \times 10^{-2}$ | $7.64 \times 10^{-2}$ | $7.83 \times 10^{-2}$ | $7.78 \times 10^{-2}$ | $8.33 \times 10^{-2}$ | $8.41 \times 10^{-2}$ |
| Astigmatism | Z2−2 | $6.50 \times 10^{-4}$ | $-2.74 \times 10^{-4}$ | $5.30 \times 10^{-5}$ | $-5.50 \times 10^{-5}$ | $-5.50 \times 10^{-5}$ | $-3.13 \times 10^{-4}$ |
| Trefoil | Z33 | $1.83 \times 10^{-4}$ | $1.90 \times 10^{-4}$ | $2.75 \times 10^{-4}$ | $3.90 \times 10^{-5}$ | $4.30 \times 10^{-5}$ | $-3.80 \times 10^{-5}$ |
| Pure Coma | Z31 | $1.35 \times 10^{-4}$ | $3.40 \times 10^{-5}$ | $-1.08 \times 10^{-4}$ | $-2.94 \times 10^{-4}$ | $-4.70 \times 10^{-5}$ | $8.60 \times 10^{-5}$ |
| Pure Coma | Z3−1 | $-2.41 \times 10^{-4}$ | $-1.50 \times 10^{-5}$ | $-3.39 \times 10^{-4}$ | $2.69 \times 10^{-4}$ | $-2.06 \times 10^{-4}$ | $1.47 \times 10^{-4}$ |
| Trefoil | Z3−3 | $-1.23 \times 10^{-4}$ | $3.60 \times 10^{-5}$ | $-1.11 \times 10^{-4}$ | $-5.40 \times 10^{-5}$ | $-1.03 \times 10^{-4}$ | $-7.30 \times 10^{-5}$ |
| Quadrofall | Z44 | $-6.40 \times 10^{-5}$ | $-3.00 \times 10^{-6}$ | $-8.00 \times 10^{-5}$ | $-1.56 \times 10^{-4}$ | $-7.50 \times 10^{-5}$ | $-3.20 \times 10^{-5}$ |
| 2−order Ast | Z42 | $2.80 \times 10^{-5}$ | $2.40 \times 10^{-5}$ | $-1.92 \times 10^{-4}$ | $-1.24 \times 10^{-4}$ | $5.40 \times 10^{-5}$ | $8.90 \times 10^{-5}$ |
| Spherical | Z40 | $9.99 \times 10^{-4}$ | $9.69 \times 10^{-4}$ | $8.48 \times 10^{-4}$ | $8.58 \times 10^{-4}$ | $1.45 \times 10^{-3}$ | $1.66 \times 10^{-3}$ |
| 2−order Ast | Z4−2 | $-2.20 \times 10^{-5}$ | $1.00 \times 10^{-5}$ | $3.20 \times 10^{-5}$ | $-4.90 \times 10^{-5}$ | $4.50 \times 10^{-5}$ | $-1.14 \times 10^{-4}$ |
| Quadrofall | Z4−4 | $-2.90 \times 10^{-5}$ | $-5.50 \times 10^{-5}$ | $1.80 \times 10^{-5}$ | $7.70 \times 10^{-5}$ | $-3.00 \times 10^{-6}$ | $-2.70 \times 10^{-5}$ |
| | | **Posterior Surface** | | | | | |
| Constant | Z00 | $1.64 \times 10^{-1}$ | $1.37 \times 10^{-1}$ | $1.64 \times 10^{-1}$ | $1.64 \times 10^{-1}$ | $1.75 \times 10^{-1}$ | $1.78 \times 10^{-1}$ |
| Tilt | Z11 | $7.05 \times 10^{-3}$ | $7.30 \times 10^{-3}$ | $2.44 \times 10^{-3}$ | $1.73 \times 10^{-3}$ | $7.10 \times 10^{-3}$ | $7.61 \times 10^{-3}$ |
| Tilt | Z1−1 | $9.83 \times 10^{-4}$ | $-4.00 \times 10^{-3}$ | $7.60 \times 10^{-4}$ | $-1.73 \times 10^{-3}$ | $4.17 \times 10^{-3}$ | $-4.14 \times 10^{-3}$ |
| Astigmatism | Z22 | $5.02 \times 10^{-3}$ | $5.45 \times 10^{-3}$ | $3.04 \times 10^{-3}$ | $3.48 \times 10^{-3}$ | $2.46 \times 10^{-3}$ | $3.04 \times 10^{-3}$ |
| Defocus | Z20 | $9.83 \times 10^{-2}$ | $9.80 \times 10^{-2}$ | $9.77 \times 10^{-2}$ | $9.76 \times 10^{-2}$ | $1.06 \times 10^{-1}$ | $1.07 \times 10^{-1}$ |
| Astigmatism | Z2−2 | $2.63 \times 10^{-4}$ | $4.36 \times 10^{-4}$ | $-7.30 \times 10^{-4}$ | $2.30 \times 10^{-4}$ | $-2.50 \times 10^{-5}$ | $2.20 \times 10^{-5}$ |
| Trefoil | Z33 | $2.54 \times 10^{-4}$ | $5.04 \times 10^{-4}$ | $2.76 \times 10^{-4}$ | $3.04 \times 10^{-4}$ | $-7.66 \times 10^{-4}$ | $-1.65 \times 10^{-4}$ |
| Pure Coma | Z31 | $1.05 \times 10^{-3}$ | $7.35 \times 10^{-4}$ | $3.19 \times 10^{-4}$ | $-6.90 \times 10^{-5}$ | $4.57 \times 10^{-4}$ | $5.21 \times 10^{-4}$ |
| Pure Coma | Z3−1 | $-1.57 \times 10^{-4}$ | $-1.64 \times 10^{-4}$ | $-3.42 \times 10^{-4}$ | $1.58 \times 10^{-4}$ | $-5.40 \times 10^{-5}$ | $1.50 \times 10^{-5}$ |
| Trefoil | Z3−3 | $-2.22 \times 10^{-4}$ | $2.23 \times 10^{-4}$ | $2.43 \times 10^{-4}$ | $-4.84 \times 10^{-4}$ | $3.28 \times 10^{-4}$ | $-5.13 \times 10^{-4}$ |
| Quadrofall | Z44 | $1.61 \times 10^{-4}$ | $4.69 \times 10^{-4}$ | $-1.50 \times 10^{-5}$ | $1.98 \times 10^{-4}$ | $-4.45 \times 10^{-4}$ | $-4.40 \times 10^{-5}$ |
| 2−order Ast | Z42 | $1.69 \times 10^{-4}$ | $-8.00 \times 10^{-5}$ | $-4.30 \times 10^{-5}$ | $7.20 \times 10^{-5}$ | $-2.29 \times 10^{-4}$ | $-5.30 \times 10^{-5}$ |
| Spherical | Z40 | $2.56 \times 10^{-3}$ | $2.74 \times 10^{-3}$ | $2.18 \times 10^{-3}$ | $2.08 \times 10^{-3}$ | $2.84 \times 10^{-3}$ | $2.99 \times 10^{-3}$ |
| 2−order Ast | Z4−2 | $-2.60 \times 10^{-5}$ | $5.20 \times 10^{-5}$ | $-7.00 \times 10^{-5}$ | $6.70 \times 10^{-5}$ | $5.80 \times 10^{-5}$ | $1.18 \times 10^{-4}$ |
| Quadrofall | Z4−4 | $-5.60 \times 10^{-5}$ | $-1.66 \times 10^{-4}$ | $-1.22 \times 10^{-4}$ | $-1.71 \times 10^{-4}$ | $-4.37 \times 10^{-4}$ | $1.81 \times 10^{-4}$ |

**Table A2.** Average Values of the Sum of the Zernike Weight Coefficients Anterior and Posterior Surfaces of the Cornea (Excluding $Zn0$).

| Type | Znm | Cnm | | |
|---|---|---|---|---|
| – | – | Healthy | Low Myopia | Medium Myopia |
| Constant | Z00 | 0 | 0 | 0 |
| Tilt | Z11 | $2.13 \times 10^{-3}$ | $7.51 \times 10^{-3}$ | $4.36 \times 10^{-3}$ |
| Tilt | Z1−1 | $5.48 \times 10^{-4}$ | $4.17 \times 10^{-3}$ | $1.91 \times 10^{-3}$ |
| Astigmatism | Z22 | $4.83 \times 10^{-3}$ | $3.37 \times 10^{-3}$ | $3.89 \times 10^{-3}$ |
| Defocus | Z20 | 0 | 0 | 0 |
| Astigmatism | Z2−2 | $-2.11 \times 10^{-3}$ | $-7.72 \times 10^{-6}$ | $-1.10 \times 10^{-3}$ |
| Trefoil | Z33 | $-3.40 \times 10^{-4}$ | $-7.53 \times 10^{-4}$ | $3.07 \times 10^{-4}$ |
| Pure Coma | Z31 | $2.50 \times 10^{-5}$ | $3.83 \times 10^{-4}$ | $-3.45 \times 10^{-4}$ |
| Pure Coma | Z3−1 | $-6.70 \times 10^{-5}$ | $-1.00 \times 10^{-4}$ | $-2.75 \times 10^{-4}$ |
| Trefoil | Z3−3 | $2.00 \times 10^{-6}$ | $3.64 \times 10^{-4}$ | $2.62 \times 10^{-4}$ |
| Quadrofall | Z44 | $-1.19 \times 10^{-4}$ | $-5.44 \times 10^{-4}$ | $3.66 \times 10^{-5}$ |
| 2−order Ast | Z42 | $-3.00 \times 10^{-5}$ | $-2.99 \times 10^{-4}$ | $-1.79 \times 10^{-4}$ |
| Spherical | Z40 | 0 | 0 | 0 |
| 2−order Ast | Z4−2 | $-2.35 \times 10^{-4}$ | $4.55 \times 10^{-5}$ | $1.63 \times 10^{-6}$ |
| Quadrofall | Z4−4 | $-2.10 \times 10^{-5}$ | $-1.49 \times 10^{-3}$ | $-1.44 \times 10^{-4}$ |

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
