# Peer review of "Simulation of the Human Myopic Eye Cornea Compensation Based on the Analysis of Aberrometric Data"

_2411-5150, 2023_

Round 1

Reviewer 1 Report

1.     The text and numbers in the Figure 1~ Figure 6 are not clearly displayed and important information cannot be seen. Especially the curves in Figure 4 and Figure 5 cannot be seen clearly.

2.     What does the symbol p in formula (1) represent?

3.     In line 345,We simulate the compensation of the coefficients of aberrations of the anterior surface of the cornea with the deviation of the total wave aberration of the cornea from the reference model by more than p%.Where p is a percentage, such as p=5%but what does p% stand for

Author Response

Point 1: The text and numbers in the Figure 1~ Figure 6 are not clearly displayed and important information cannot be seen. Especially the curves in Figure 4 and Figure 5 cannot be seen clearly.

Response 1: Thanks for your remark. Figures 1-3, 6 are corrected. Figure 4 is a screenshot from the program, it is fully scaled. Figure 5 adds a scale of values and a legend in the description.

Point 2: What does the symbol p in formula (1) represent?

Response 2: The symbol p in the formula (1) means the sum index and changes as indicated in the formula (1) from 0 to (n-m)/2.

Point 3: In line 345, “We simulate the compensation of the coefficients of aberrations of the anterior surface of the cornea with the deviation of the total wave aberration of the cornea from the reference model by more than p%.” Where p is a percentage, such as p=5%,but what does p% stand for?

Response 3: Thanks for your question! To avoid confusion with the index p in Eq. (1), we have replaced the deviation by symbol s. Indeed, the parameter s is entered in %. It means the threshold value at which the aberration Znm is compensated if the weight coefficient corresponding to it deviates in the myopic eye from the corresponding weight coefficient of the healthy eye by more than s%.

Reviewer 2 Report

The paper presents an innovative approach for human myopic eye cornea compensation based on aberrometric data in contrast to a large number of well-established methods associated with different wavefront aberration sensors.

The manuscript is clearly explaining the method for Zernike polynomial coefficients determination in case of the anterior and posterior surfaces of the eye cornea. These results are relevant for the field and presented in a well-structured manner.

The paper has 61 relevant cited references, 21 are recent publications (within the last 5 years). The number of self-citation papers is high (17) due to the fact that the authors have been intensively studied the field since 1990 (their pioneer work regarding a method for detecting wavefront aberrations can be found published in [22-24]).

However, the reference list could be improved with recent papers regarding the development of supporting technologies for new types of wavefront sensors. For example, at line 125 one recent relevant work on multilevel diffractive axicons could be added (DOE not being addressed on SLM bring the advantage of compact optical systems, moreover multilevel axicons by comparison to binary axicons have higher conversion efficiency) Nanomaterials 2023, 13(3), 579; https://doi.org/10.3390/nano13030579

The results from this paper are expected to have high social impact regarding the quality of vision aspects. Also, one can anticipate the paper’s potential to improve the ophthalmological diagnosis and treatment after medical examinations of the cornea in the human eye, in particular in the case of people with myopia (precise vision correction by compensating the aberrations).

Add color bar for Figure 5a, Figure 5b and legend for the graphs in the case of Figure 5c.

The legend of Figure 6 should be translated into English.

Line 408, “the total wave aberration of each Zernike polynomial decreases after ophthalmic correction”. It seems from the graphic that the absolute values of the coefficients decrease.

Line 507 The phrase from line 507 is not clear.

The paper can be accepted for publication, with minor revisions, mentioned above.

Author Response

Point 1: The paper presents an innovative approach for human myopic eye cornea compensation based on aberrometric data in contrast to a large number of well-established methods associated with different wavefront aberration sensors.

The manuscript is clearly explaining the method for Zernike polynomial coefficients determination in case of the anterior and posterior surfaces of the eye cornea. These results are relevant for the field and presented in a well-structured manner.

Response 1: Thank you for your kind words!

Point 2: The paper has 61 relevant cited references, 21 are recent publications (within the last 5 years). The number of self-citation papers is high (17) due to the fact that the authors have been intensively studied the field since 1990 (their pioneer work regarding a method for detecting wavefront aberrations can be found published in [22-24]).

Response 2: Thank you for your mark! At the request of the publisher, we have reduced the number of self-citations.

Point 3: However, the reference list could be improved with recent papers regarding the development of supporting technologies for new types of wavefront sensors. For example, at line 125 one recent relevant work on multilevel diffractive axicons could be added (DOE not being addressed on SLM bring the advantage of compact optical systems, moreover multilevel axicons by comparison to binary axicons have higher conversion efficiency) Nanomaterials 2023, 13(3), 579; https://doi.org/10.3390/nano13030579

Response 3:  Thanks for your recommendation. We've added a relevant reference to this work in the introduction.

Point 4: The results from this paper are expected to have high social impact regarding the quality of vision aspects. Also, one can anticipate the paper’s potential to improve the ophthalmological diagnosis and treatment after medical examinations of the cornea in the human eye, in particular in the case of people with myopia (precise vision correction by compensating the aberrations).

Response 4: Thank you for your appreciation of our results!

Point 5: Add color bar for Figure 5a, Figure 5b and legend for the graphs in the case of Figure 5c.

Response 5: Thank you for your comment. A scale of values and a legend in the description added to Figure 5.

Point 6: The legend of Figure 6 should be translated into English.

Response 6: In Figure 6, the scale of values is scaled and the legend is translated into English.

Point 7: Line 408, “the total wave aberration of each Zernike polynomial decreases after ophthalmic correction”. It seems from the graphic that the absolute values of the coefficients decrease.

Response 7: Figure 6 shows that after correction, the Zernike weight coefficients decrease on average, which should lead to an improvement in the quality of the PSF and the patient's vision. This is confirmed by the distribution of intensity on the PSF in Table 4 and by reducing the RMS of the obtained PSF relative to the ideal one (Airy spot). We have added this explanation to the text.

Point 8: Line 507 The phrase from line 507 is not clear.

Response 8: We have removed this sentence when shortening the text in the Conclusions section.

Point 9: The paper can be accepted for publication, with minor revisions, mentioned above.

Response 9: Thank you for your final decision!

Reviewer 3 Report

Minor concerns

#1 The clinical implications of this work should be described, which could change with respect to current techniques.

#2 The title is very general.

#3 The limitations of the study should be included in the discussion and explained.

#4 The conclusions of the study are very long, some of them should be included in the discussion and be more specific, and that they respond to a more specific title.

Author Response

Point 1: The clinical implications of this work should be described, which could change with respect to current techniques.

Response 1: Thank you for your remark. We have supplemented the Discussion section with a description of the clinical implications of this work in the context of method and technology developments.

Point 2: The title is very general.

Response 2: Thank you for your comment, but allow us to disagree with you. The title clearly indicates what type of eye disease we are considering and what optical surface of the eye. By "aberrometric data" we mean the standard expansion of wavefront aberrations in terms of Zernike polynomials, which is the OSA and ANSI standard.

Point 3: The limitations of the study should be included in the discussion and explained.

Response 3: We took into account your remark and determined the limitations of the study. We explained how they affect the results of the study and suggested directions for future research in the Discussion section.

Point 4: The conclusions of the study are very long, some of them should be included in the discussion and be more specific, and that they respond to a more specific title.

Response 4: The Discussion section is supplemented with some general inference. The Conclusions section has been made more concise with the concrete results of this study.
